# An extension of the logistic function to account for nonstationary drought losses

Tongtiegang Zhao[1], Zecong Chen[1], Yongyong Zhang[2], Bingyao Zhang[3], and Yu Li[3]

[1] Southern Marine Science and Engineering Guangdong Laboratory (Zhuhai), School of Civil Engineering, Sun Yat-Sen University, Guangzhou 510275, China
[2] Key Laboratory of Water Cycle and Related Land Surface Processes, Institute of Geographic Sciences and Natural Resources Research, Chinese Academy of Sciences, Beijing 100101, China
[3] School of Hydraulic Engineering, Dalian University of Technology, Dalian 116024, Liaoning, China

*Corresponding author:* Tongtiegang Zhao (zhaottg@mail.sysu.edu.cn) and Zecong Chen (chenzc8@mail2.sysu.edu.cn)

**Highlights:**

1. The drought-affected population in mainland China exhibits remarkable correlation not only with standard precipitation index, but also with time;

2. The nonstationary drought losses are effectively characterized by incorporating time into the parameters of the classic logistic function;

3. The nonstationary intensity loss functions built upon the logistic function are demonstrated to be a useful tool for drought impact assessment.

**Abstract:** While the stationary intensity loss function is fundamental to drought impact assessment, the relationship between drought loss and intensity can be nonstationary, i.e., changing as time progresses, owing to socio-economic developments. This paper addresses this critical gap by modelling nonstationary drought losses. Specifically, the time is explicitly formulated by linear and quadratic functions and then incorporated into the magnitude, shape and location parameters of the logistic function to derive in total six nonstationary intensity loss functions. To examine the effectiveness, a case study is designed for the drought-affected population by province in mainland China during the period from 2006 to 2023. The results highlight the existence of nonstationarity in that the drought-affected population exhibits significant correlation not only with standard precipitation index but also with time. The proposed nonstationary intensity loss functions are shown to outperform not only the classic logistic function but also the linear regression. They present effective characterizations of observed drought loss in different ways: 1) the nonstationary function with the flexible magnitude parameter fits the data by adjusting the maximum drought loss by year; 2) the nonstationary function with the flexible shape parameter works by modifying the growth rate of drought loss with intensity; and 3) the nonstationary function with the flexible location parameter acts by shifting the response curves along the axis by year. Among the nonstationary logistic functions, the function incorporating the linear function of time into the magnitude parameter generally outperform the others in terms of high coefficient of determination, low Bayesian information criterion and explicit physical meaning. Taken together, the nonstationary intensity loss functions developed in this paper can serve as an effective tool for drought management.

**Short summary:** The classic logistic function characterizes the stationary relationship between drought loss and intensity. This paper accounts for time in the magnitude, shape and location parameters of the logistic function and derives nonstationary intensity loss functions. A case study is designed to test the functions for drought-affected population by province in mainland China from 2006 to 2023. Overall, the nonstationary intensity loss functions are shown to be a useful tool for drought management.

## 1 Introduction

Droughts are one of the most destructive natural hazards (Baez-Villanueva et al., 2024; Van Dijk et al., 2013; Zhang et al., 2022). In general, there exist meteorological, hydrological, agricultural and socio-economic droughts (Mishra and Singh, 2010). Originating from precipitation deficits and high atmospheric evaporative demands, droughts propagate through hydrological processes and eventually impair human beings and natural ecosystems (Gao et al., 2024a; Liu et al., 2024; Zhao et al., 2024a). From 2001 to 2009, the "Millennium Drought" in Southeast Australia amplified median rainfall reduction by up to 4 times in streamflow and reduced irrigated rice and cotton production respectively by 99% and 84% (Van Dijk et al., 2013). The 2012 summertime drought arrived at the Central Great Plains in North America without early warning and caused more than US$30 billion of economic losses (Hoerling et al., 2014; Yuan et al., 2023). The 2021/22 drought event made 76.2% of the Euro-Mediterranean region under mild drought, 61.4% under moderate drought and 39.4% under severe drought (Garrido-Perez et al., 2024). Under climate change, droughts are expected to not only increase worldwide (Dai, 2011) but also intensify more rapidly (Yuan et al., 2023).

Socio-economic losses are an integral part of droughts in environment management (AghaKouchak et al., 2021; Hoerling et al., 2014; Van Dijk et al., 2013). Although there exist extensive studies on hydroclimatic processes associated with droughts (Entekhabi, 2023; Mishra and Singh, 2010; Wang et al., 2023b; Yang et al., 2024; Zhang et al., 2021), far less attention is paid to socio-economic impacts of droughts (AghaKouchak et al., 2021; Apurv and Cai, 2021; Su et al., 2018). One possible cause is the lack of socio-economic data on droughts (Su et al., 2018; Yang et al., 2024). On the one hand, in situ observations, satellite remote sensing and earth system models generate a vast amount of hydroclimatic data (Hersbach et al., 2020; Pradhan et al., 2022; Zhang et al., 2024, 2021; Zhao et al., 2024b). Plenty of spatial-temporal data facilitate drought investigations at catchment, regional, continental and global scales and in pentad, monthly, seasonal and annual time steps (Gao et al., 2024b; Ma et al., 2022; Wang et al., 2023a). On the other hand, there are limited data on socio-economic losses due to droughts (AghaKouchak et al., 2021). Usually, drought losses have to be collected from statistical yearbooks issued by local and central governments and from survey reports provided by international organizations and commercial services (Chen et al., 2015; Hou et al., 2019).

The intensity loss function, which is also described as exposure-response curve and dose–response relationship, plays a critical part in disaster risk management (AghaKouchak et al., 2021; Qiu et al., 2023; West et al., 2019). The classic logistic function is effective in characterizing the growth of socio-economic loss with drought intensity (Chen et al., 2015; Hou et al., 2019; Todisco et al., 2013). Meanwhile, the relationship between socio-economic loss and drought intensity can be nonstationary, i.e., temporally changing, considering that economic growth can increase the exposure to droughts and that infrastructure developments can increase the resilience to droughts (Apurv and Cai, 2021; Haile et al., 2020; Long et al., 2020). In this paper, we build three non-stationary functions upon the magnitude, shape and location parameters of the classic logistic function that represents a stationary intensity loss function. As will be illustrated in the methods and results, the proposed functions tend to

capture the non-stationary characteristics of drought-affected population in mainland China. The effects of drought intensity and time on population in different provinces are effectively characterized.

## 2 Methods

### 2.1 Intensity loss function

Drought indices are essential for drought impact assessment (Montanari et al., 2023; Todisco et al., 2013; West et al., 2019). Among the popular indices are the standardized precipitation index (SPI), the Palmer drought severity index (PDSI), the standardized precipitation evapotranspiration index (SPEI) and the standardized runoff index (SRI) (AghaKouchak et al., 2021; Apurv and Cai, 2021; Zhao et al., 2024b). The intensity is derived from drought indices (Hao et al., 2017; Mishra and Singh, 2010; Su et al., 2018). Since 0 is both the mean and median values of the standard normal distribution, the extent to which drought indices falling below 0 indicates the degree of dryness. Thresholds can be employed to identify drought events (Wang et al., 2023b). For example, (–0.99, 0] is near normal, (–1.49, –1.00] is moderately dry, (–1.99, –1.50] is severely dry and (–∞, –2.00] is extremely dry. Therefore, drought events can be defined by the combinations of multiple indices, e.g., by $SPI \leq -1.0$, $PDSI \leq -2.0$ and $SPEI \leq -1.0$ (Su et al., 2018).

Denoting the drought intensity as $I$, the intensity loss function is formulated as:

$$L = f(I) \tag{1}$$

in which $L$ is the drought loss corresponding to the intensity $I$. Empirically, there are four important characteristics of $f(I)$: 1) there is minimal loss when there is minimal $I$; 2) there is maximal loss when there is maximal $I$; 3) $f(I)$ is a monotonically increasing function, i.e., drought loss increases with drought intensity; and 4) drought loss grows slowly with $I$ initially, rises rapidly as $I$ increases and then slows down until maturity.

The above four characteristics can mathematically be formulated by the renowned logistic function (Chen et al., 2015; Jonkman et al., 2008; Kucharavy and De Guio, 2011):

$$L(I) = \frac{A}{1 + e^{-k(I-c)}} \tag{2}$$

in which there are three parameters: 1) the magnitude parameter $A$ representing the maximum drought loss; 2) the shape parameter $k$ controlling the growth rate of $L$ with $I$; and 3) the location parameter $c$ indicating the point at which the saturation begins.

As to drought indices that represent the intensity, they can be derived from the target hydroclimatic variable's cumulative distribution function (CDF) and the inverse CDF of the standard normal distribution (Hao et al., 2017; Mishra and Singh, 2010; Montanari et al., 2023; Zhang et al., 2024; Zhao et al., 2024b). For example, the SPI is calculated as:

$$SPI_t = CDF_{N(0,1)}^{-1}\left(CDF_p(p_t)\right) \tag{3}$$

in which $SPI_t$ in period $t$, which follows the standard normal distribution, is derived from precipitation amount $p_t$ in period $t$.
There are two steps: firstly, $p_t$ is converted into a standard uniform variate between 0 and 1 by its CDF, i.e., $CDF_p(\cdot)$; and secondly, the standard uniform variate is converted into the standard normal variate $SPI_t$ by the inverse CDF of $N(0,1^2)$, i.e., $CDF_{N(0,1)}^{-1}(\cdot)$. Similarly, the SPEI is derived from the difference between precipitation and potential evapotranspiration (Baez-Villanueva et al., 2024). Furthermore, the self-calibrating PDSI (scPDSI) takes into account evapotranspiration, recharge, runoff and loss so as to report dry conditions with frequencies that would be expected for rare conditions (Wells et al., 2004).


## 2.2 Formulation of the logistic function

There is an inverse relationship between drought intensity and drought indices like SPI, SPEI and scPDSI. It is because the extent of dryness is generally characterized by how negative drought indices are (Haile et al., 2020; Liu et al., 2024; Zhang et al., 2021). That is, the more intensive dryness, the more negative drought indices. Taking the SPI as an indicator of drought
intensity, the logistic function is modified by removing the negative sign in front of $k$:

$$L(SPI) = \frac{A}{1 + e^{k(SPI-c)}} \tag{4}$$

The ranges of the three parameters can be predetermined in accordance with the physical meanings of the parameters:
First of all,

$$A > 0 \tag{5}$$

which means that drought loss is always above zero.
Secondly,

$$k > 0 \tag{6}$$

which means that as $SPI$ increases from $-\infty$ to $+\infty$, the denominator in Eq. (6) increases and leads to the reduction of drought loss. Eventually, the increasing denominator makes drought loss approach zero when SPI is large enough. On the other hand, it is noted that the loss would turn to increase with SPI when $k$ is negative.
Thirdly,

$$-\infty < c < +\infty \tag{7}$$

which means that the value of $c$ depends on the case under investigation and can change freely.


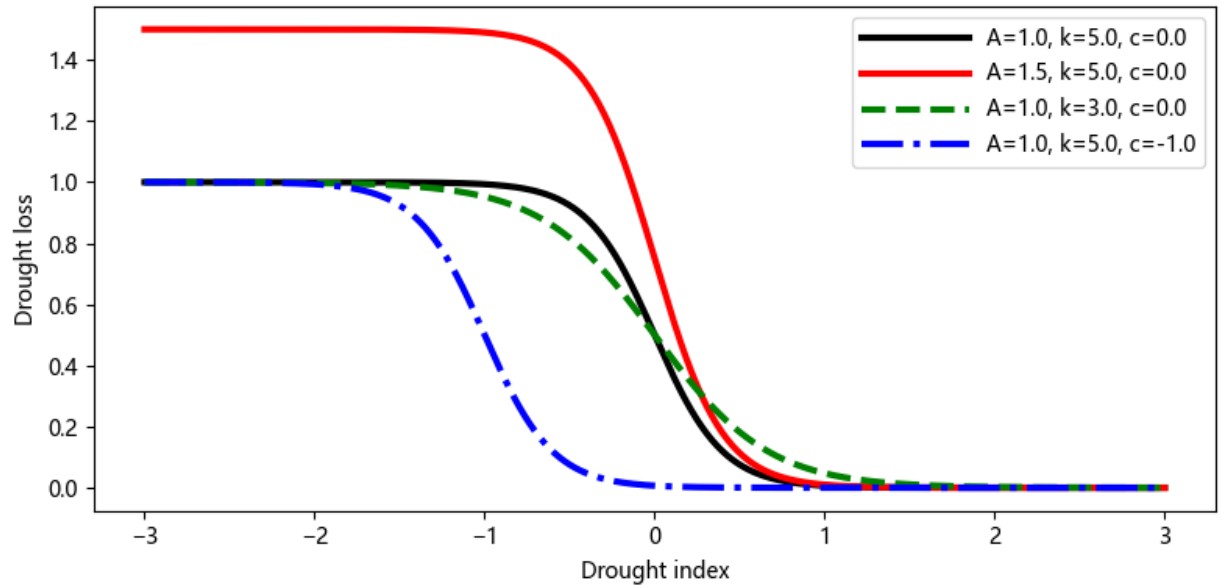

**Figure 1. An illustrative example of the logistic function under four sets of parameters.**

An illustrative example of the logistic function in Eq. (4) is presented in Figure 1. The result under the basic parameter set of
($A$=1.0, $k$=5.0, $c$=0.0) is marked in black. There are three one-factor-at-a-time experiments (Chen and Zhao, 2020). Firstly, the value of $A$ is increased to 1.5. As is shown by the red line, the maximum drought loss evidently increases but the shape of the line stays the same. Secondly, the value of $k$ is reduced to 3.0. As is shown by the green line, the shape of the line becomes flatter but the maximum loss remains the same. Thirdly, the value of $c$ is decreased to –1.0. As is shown by the blue line, the curve is shifted to the left as a whole while both the maximum loss and shape do not change.


### 2.3 Stationary and non-stationary formulations

There are socio-economic factors contributing to temporal changes, i.e., nonstationarity, of the intensity loss function (AghaKouchak et al., 2021; Chiang et al., 2021; Long et al., 2020). Firstly, the exposure to drought can increase with time owing to increases of population, accumulations of wealth and developments of infrastructure. Secondly, the vulnerability
under a given level of drought intensity may decrease with time considering engineering measures, such as constructions of water storage reservoirs and inter-basin water diversion projects. Thirdly, the resilience to drought can be improved by drought management measures such as sub-seasonal to seasonal hydroclimatic forecasting and forecast-informed reservoir operation. In general, the relationship between drought loss and intensity tends to evolve as time progresses due to socio-economic developments and deployments of engineering and non-engineering drought-coping strategies (Hou et al., 2019; Jonkman et
al., 2008; Su et al., 2018).

Without considering temporal changes, there is a stationary logistic function $L_{A0k0c0}(\cdot)$:

$$L_{A0k0c0}(SPI_t) = \frac{A_0}{1 + e^{k_0(SPI_t - c_0)}} \qquad (8)$$

To account for temporal change, the linear function that takes time $t$ as an explanatory variable (Cheng et al., 2014; Xiong et al., 2015) can be formulated for the parameters $A$, $k$ and $c$:

$$\begin{cases} A_t = A_0 + A_1 \times t \\ k_t = k_0 + k_1 \times t \\ c_t = c_0 + c_1 \times t \end{cases} \qquad (9)$$

in which $A_0$, $k_0$ and $c_0$ are the intercepts while $A_1$, $k_1$ and $c_1$ are the slopes. The incorporation of Eq. (9) into Eq. (8) yields the following three equations:

$$\begin{cases} L_{A1k0c0}(SPI_t) = \dfrac{A_0 + A_1 \times t}{1 + e^{k_0(SPI_t - c_0)}} \\[2ex] L_{A0k1c0}(SPI_t) = \dfrac{A_0}{1 + e^{(k_0 + k_1 \times t) \times (SPI_t - c_0)}} \\[2ex] L_{A0k0c1}(SPI_t) = \dfrac{A_0}{1 + e^{k_0(SPI_t - (c_0 + c_1 \times t))}} \end{cases} \qquad (10)$$

in which the logistic functions $L_{A0k1c0}(\cdot)$, $L_{A0k1c0}(SPI_t)$ and $L_{A0k0c1}(SPI_t)$ respectively have nonstationary magnitude, shape and location parameters.

Furthermore, the quadratic function can be used to accommodate possibly nonlinear changes:

$$\begin{cases} A_t = A_0 + A_1 \times t + A_2 \times t^2 \\ k_t = k_0 + k_1 \times t + k_2 \times t^2 \\ c_t = c_0 + c_1 \times t + c_2 \times t^2 \end{cases} \qquad (11)$$

The incorporation of Eq. (11) into Eq. (8) yields another three equations:

$$\begin{cases} L_{A2k0c0}(SPI_t) = \dfrac{A_0 + A_1 \times t + A_2 \times t^2}{1 + e^{k_0(SPI_t - c_0)}} \\[2ex] L_{A0k2c0}(SPI_t) = \dfrac{A_0}{1 + e^{(k_0 + k_1 \times t + k_2 \times t^2) \times (SPI_t - c_0)}} \\[2ex] L_{A0k0c2}(SPI_t) = \dfrac{A_0}{1 + e^{k_0(SPI_t - (c_0 + c_1 \times t + c_2 \times t^2))}} \end{cases} \qquad (12)$$

In Eq. (8), Eq. (10) and Eq. (12), the subscripts "Ax", "kx" and "cx" are respectively for the magnitude, shape and location parameters. As to "x", the values 0, 1 and 2 respectively indicate the non-involvement of time, the linear function of time and the quadratic function of time. As a result, the logistic function is non-stationary when x is 1 or 2. For example, $L_{A1k0c0}(SPI_t)$ represents the nonstationary logistic function involving the linear function of time for the magnitude parameter.

The fitting of the stationary and nonstationary functions is considered to be a nonlinear least-squares problem by searching for
the set of parameters that minimize the sum of squares of residuals. It is performed by the curve_fit function in the SciPy optimization toolbox (Virtanen et al., 2020).

## 3 Case study

### 3.1 Data description

The drought loss data is sourced from the Ministry of Water Resources (MWR) of China. The MWR has published by year "Bulletin of Flood and Drought Disaster in China" since 2006. The name of the bulletin was changed to "China Flood and Drought Disaster Prevention Bulletin" in 2019. By collating floods and droughts reported by provincial governments and river basin commissions, the MWR has presented in the bulletin major events of droughts and floods across the 31 provinces in mainland China. As to droughts and floods in each province, the bulletin provides by year the quantitative socio-economic
losses, contingency plans and retrospective analysis of prevention and control measures.

The attention is paid to the drought-affected population, which represents the number of individuals suffering from droughts as recorded in official reports. In Figure 2 are the multi-annual mean drought-affected population, maximum annual drought-affected population, mean annual precipitation and total population. From Figures 2a and 2b, it can be observed that provinces in Southwest China, including Yunnan, Guizhou and Sichuan Provinces, tend to have the largest population suffering from
droughts. Particularly in 2010, 9.65 million people in Yunnan Province and 7.57 million people in Guizhou Province were struck by a record-breaking drought event induced by the persistently positive Madden-Julian Oscillation (Lü et al., 2012). On the other hand, it can be seen from Figures 2c and 2d that there is neither low precipitation nor large population in Southwest China. In general, the large drought-affected population in Yunnan and Sichuan Provinces is attributed to the Karst landscape, which features small storage capacity, high infiltration rate and fast groundwater flow (Wan et al., 2016).


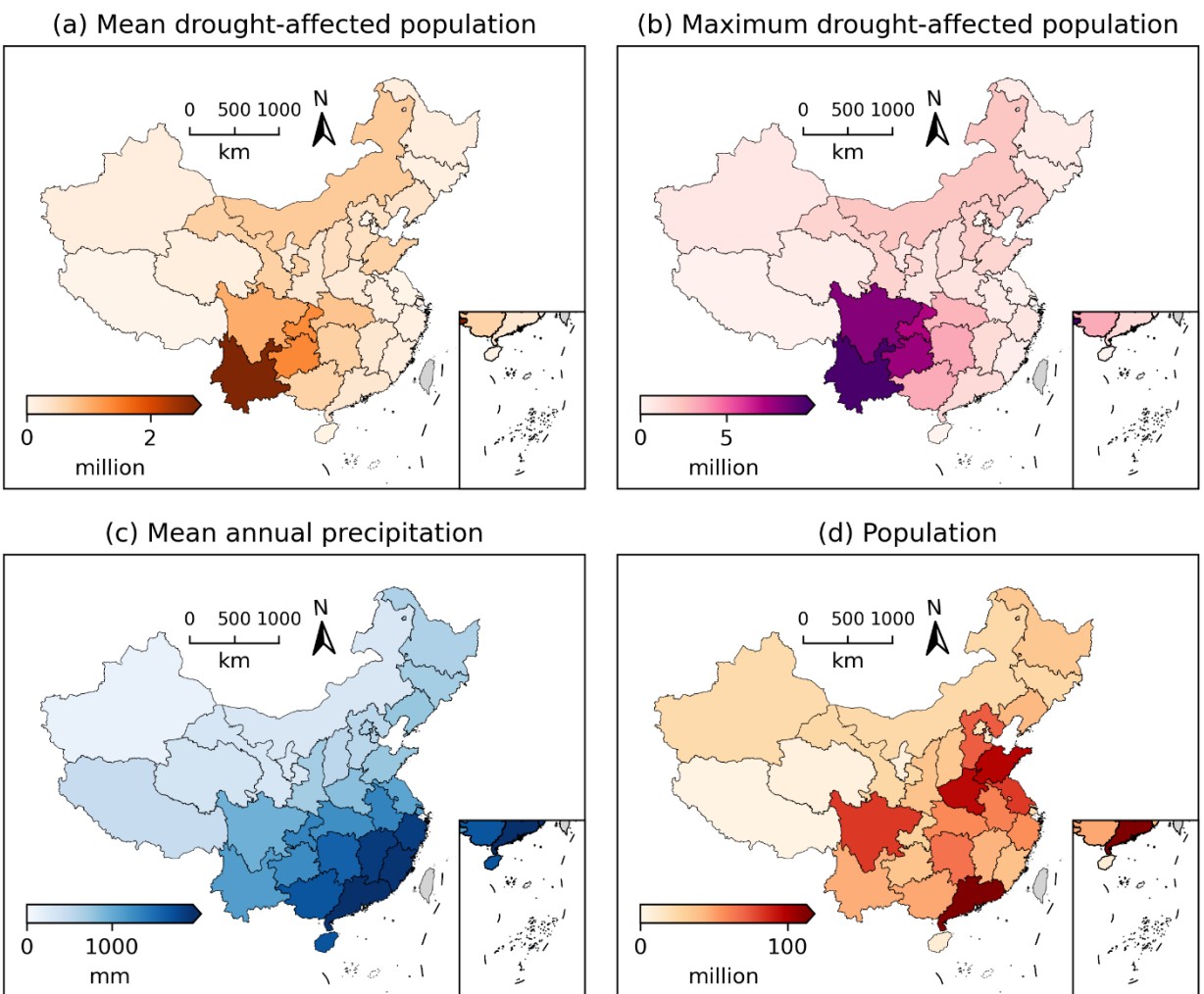

**Figure 2. Spatial plots of (a) mean annual drought-affected population, (b) maximum annual drought-affected population, (c) mean annual precipitation and (d) population by province in mainland China.**

The precipitation data used for the calculation of SPI is obtained from the Climate Hazards Group InfraRed Precipitation with Station data (CHIRPS) (Funk et al., 2015). The intersection operation is performed to use provincial polygons to extract spatially averaged precipitation from the raster CHIRPS precipitation field. To better characterize the climatological distribution of precipitation, time series of annual precipitation are extracted by province for the period from 1981 to 2023. The 43 years' annual precipitation is firstly converted into CDF by the Weibull's plotting position (Ye et al., 2018) and then converted into SPI by the inverse CDF of $N(0,1^2)$. Then, the SPI in the years from 2006 to 2023 is used in the fitting of the logistic functions.

Alongside the SPI, the SPEI data is obtained from the SPEIbase (Beguerá et al., 2024). Specifically, the SPEI-12 in December is selected to represent the annual drought condition as the loss is at the annual timescale; and the intersection operation is performed to use provincial polygons to extract spatially averaged SPEI. Furthermore, the scPDSI is sourced from the Climate Research Unit (CRU) (Barichivich et al., 2024). As the scPDSI is monthly, the values across the 12 months within a year are averaged before taking the spatial average of each province.

## 3.2 Model evaluation

The coefficient of determination, i.e., $R^2$, is evaluated for the stationary logistic function (Eq. 8) and the six types of nonstationary logistic functions (Eqs. 10 and 12). That is, the sum of squares of residuals for the estimations provided by the functions is compared to the baseline sum of squares of residuals for the mean value. As a result, $R^2$ represents the ratio of total variation of the drought loss that is explained:

$$R^2 = 1 - \frac{\sum_t \left(L_t - \hat{L}_t\right)^2}{\sum_t (L_t - \bar{L})^2} \tag{13}$$

in which $L_t$ is the drought loss in year $t$, $\hat{L}_t$ is the loss estimated by the function under investigation and $\bar{L}$ is the mean value of all $L_t$.

The number of parameters plays a critical part in statistical modelling. That is, more parameters facilitate more flexible fitting of observed data but in the meantime, are more prone to overfitting (Neath and Cavanaugh, 2012). There are 3 parameters for the stationary logistic function, 4 parameters for the non-stationary logistic functions with the linear function and 5 parameters for the non-stationary logistic functions with the quadratic function:

$$\begin{cases} n_{A0k0c0} = 3 \\ n_{A1k0c0} = 4 \\ n_{A0k1c1} = 4 \\ n_{A0k0c1} = 4 \\ n_{A2k0c0} = 5 \\ n_{A0k2c0} = 5 \\ n_{A0k0c2} = 5 \end{cases} \tag{14}$$

The Bayesian information criterion (BIC) takes into account both the sum of squares of residuals and the number of parameters (Neath and Cavanaugh, 2012):

$$BIC = T \times \ln\left(\frac{\sum_t \left(L_t - \hat{L}_t\right)^2}{T}\right) + n \times \ln(T) \tag{15}$$

in which $\ln(\cdot)$ is the natural logarithmic function, $T$ is the number of observations and $n$ is the number of parameters. BIC is negatively oriented, meaning that a lower value indicates a better fit. As a result, both larger sum of squares of residuals and more parameters are penalized by the BIC.

## 4 Results

### 4.1 Correlation analysis

The Pearson's correlation coefficient between drought-affected population and time as well as SPI is illustrated by bar plots in Figure 3. There are in total 31 provincial administrative regions in mainland China. Beijing, Tianjin, Shanghai and Xizang are not considered since they are free from drought-affected population in most years. This outcome is mainly due to ample water availability and water supply facilities (Long et al., 2020; Sun et al., 2021). For the other 27 provincial administrative regions, it can be observed from Figure 3a that the correlation coefficient between drought-affected population and time is mostly significantly negative. Meanwhile, it is slightly positive in Guangdong and Fujian Provinces although not significant. The implication is that the drought-affected population mostly exhibits a decreasing trend as time progresses and sometimes shows an increasing trend. From Figure 3b, it is seen that the correlation coefficient between drought-affected population and SPI is in general significantly negative. This result suggests that drought-affected population tends to decrease as the amount of precipitation increases. Overall, the correlation coefficients in Figure 3 point out that it is reasonable to use both time and SPI as explanatory variables of drought-affected population.

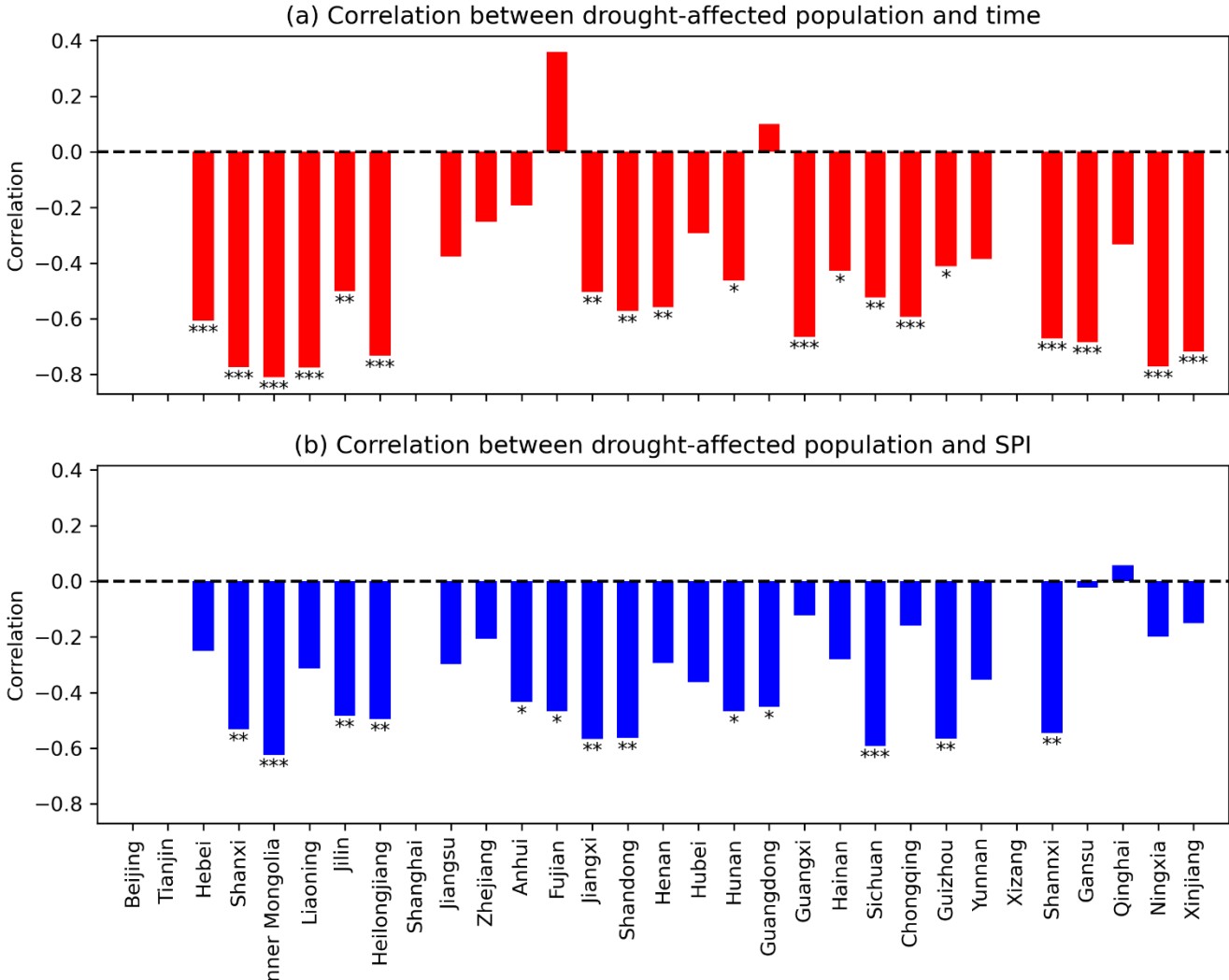

Figure 3. Correlation coefficient between drought-affected population and (a) time as well as (b) SPI by province. Alongside the bars are *, ** and *** respectively indicating the significance at the levels of 0.10, 0.05 and 0.01. Bars without * imply non-significant correlation coefficients.

The drought-affected population is plotted against time and SPI for Yunnan Province in Figure 4 due to its remarkable mean annual drought-affected population (Wan et al., 2016) and for Guangdong Province in Figure 5 due to its economic importance (Shao et al., 2020). The scatter plots on the left-hand side of the two figures imply the complexity of drought impact assessment. That is, owing to socio-economic developments, the drought-affected population can decrease or increase as time progresses (Apurv and Cai, 2021; Haile et al., 2020; Long et al., 2020). In the meantime, the scatter plots on the right-hand side suggest that the increase of precipitation amounts effectively reduces the number of population suffering from droughts (AghaKouchak et al., 2021; Qiu et al., 2023; West et al., 2019).

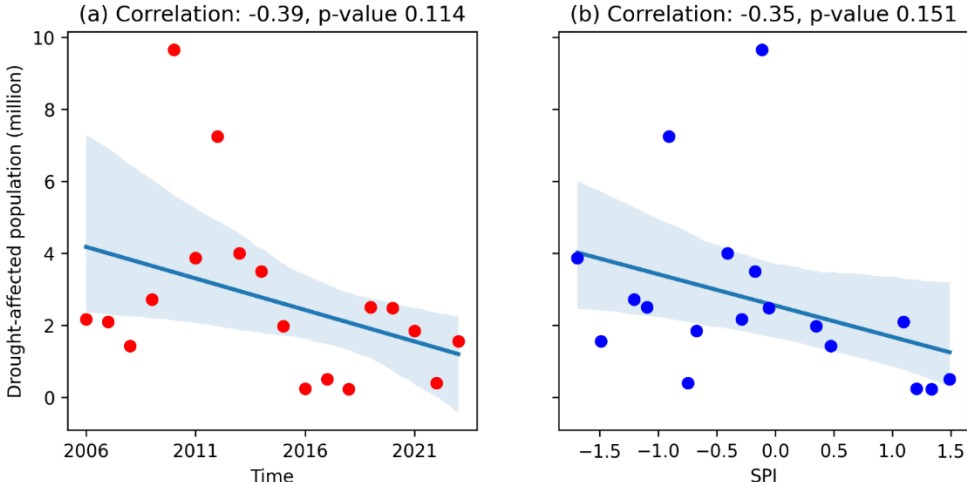

**Figure 4. Scatter plots of drought-affected population against (a) time and (b) SPI in Yunnan Province.**

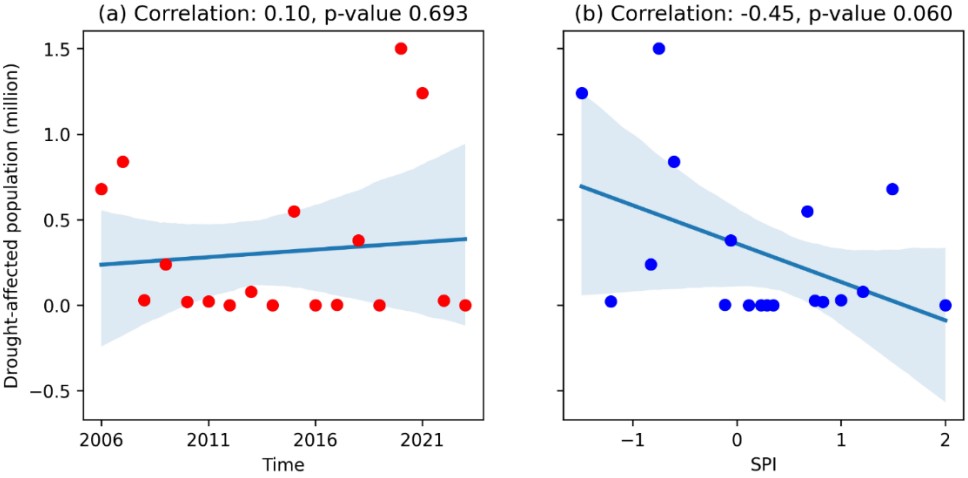

**Figure 5. As for Figure 4, but for Guangdong Province.**

## 4.2 Decreasing drought-affected population

The stationary logistic function directly relates the drought-affected population to SPI (Figure 6), while the nonstationary logistic functions account for the dependency of drought-affected population on both SPI and time (Figures 7 and 8).

In Figure 6, it is shown that the mean drought-affected population is about 4 million. Yet, the maximum was up to 10 million in the year of 2010. Furthermore, the data point with the maximum drought-affected population happened to be with a SPI that

is around 0, which is owing to that drought conditions depend not only on precipitation, but also on evapotranspiration, water storage and other hydroclimatic factors (Su et al., 2018; Yin et al., 2022a, b). In general, it is hard for the stationary logistic

function A0k0c0 to capture the data points with lower SPI but smaller drought-affected population.

In Figure 7, the nonstationary logistic functions A1k0c0, A0k1c0 and A0k0c1 are visualized by the surface and wireframe plots. While the correlation between drought-affected population and time tends to be negative in Yunnan Province, it is observed that the nonstationary functions tend to capture not only the decrease of drought-affected population with SPI, but also the decrease of drought-affected population with time. Since the year with the maximum drought-affected population is

260 in the early part of the study period, there is a remarkable increase in $R^2$. The three functions perform differently in capturing the observed data points:

1) The flexible magnitude parameter in A1k0c0 tends to fit the observed data by reducing the maximum drought loss by year (Figure 6a). As can be seen from the wireframe plot, the maximum drought loss evidently reduces from 2006 to 2023 while the shape and location of the curves remain the same.

2) The flexible shape parameter in A0k1c0 fits the observed data by changing the response surface, as shown in Figure 6b. Although it exhibits the highest $R^2$ and the lowest BIC, the fitted drought-affected population is shown to counterintuitively increase with SPI in 2021, 2022 and 2023. That is, more people could be subject to drought as precipitation increases in these three years. This wrong outcome is owing to the flexibility of the shape parameter. Specifically, the value of the shape parameter can be forced by the trend term to turn from positive to negative as time

progresses. When the shape parameter is negative, the estimated drought impact would increase with the precipitation amount.

3) The flexible location parameter in A0k0c1 tends to fit the observation data by shifting the response curves by year, as shown in Figure 6c. Due to that the maximum drought-affected population is fixed from 2006 to 2023, it is observed that the maximum affected population in 2010 is not effectively captured.

In Figure 8, the nonstationary logistic functions A2k0c0, A0k2c0 and A0k0c2 are also visualized by the surface and wireframe plots. Although the quadratic function leads to some improvements in $R^2$, the improvements are at the cost of the physical meaning of the results. From Figure 8a, it is observed that under a given SPI that is below 0, the drought-affected population initially increases but then decreases with time. From Figure 8b, it is observed that the response surface exhibits a complex shape that can be due to the fitting of sample-specific noise. The implication is that the data points are too limited to facilitate

the fitting of quadratic function in A2k0c0, A0k2c0 and A0k0c2.

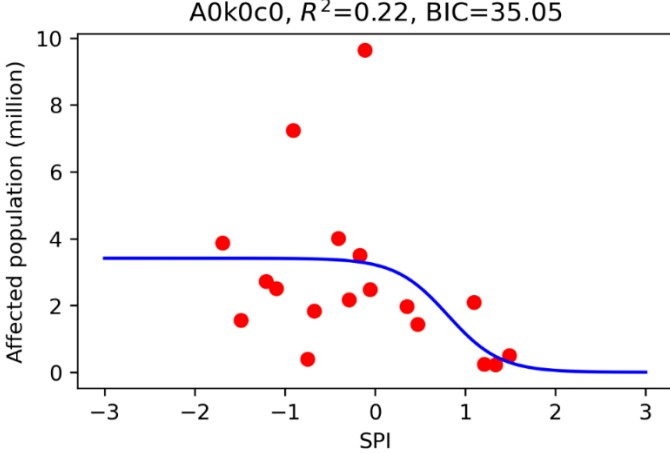

**Figure 6. Illustration of the stationary logistic function A0k0c0 fitting the relationship between SPI and drought-affected population for Yunnan Province.**

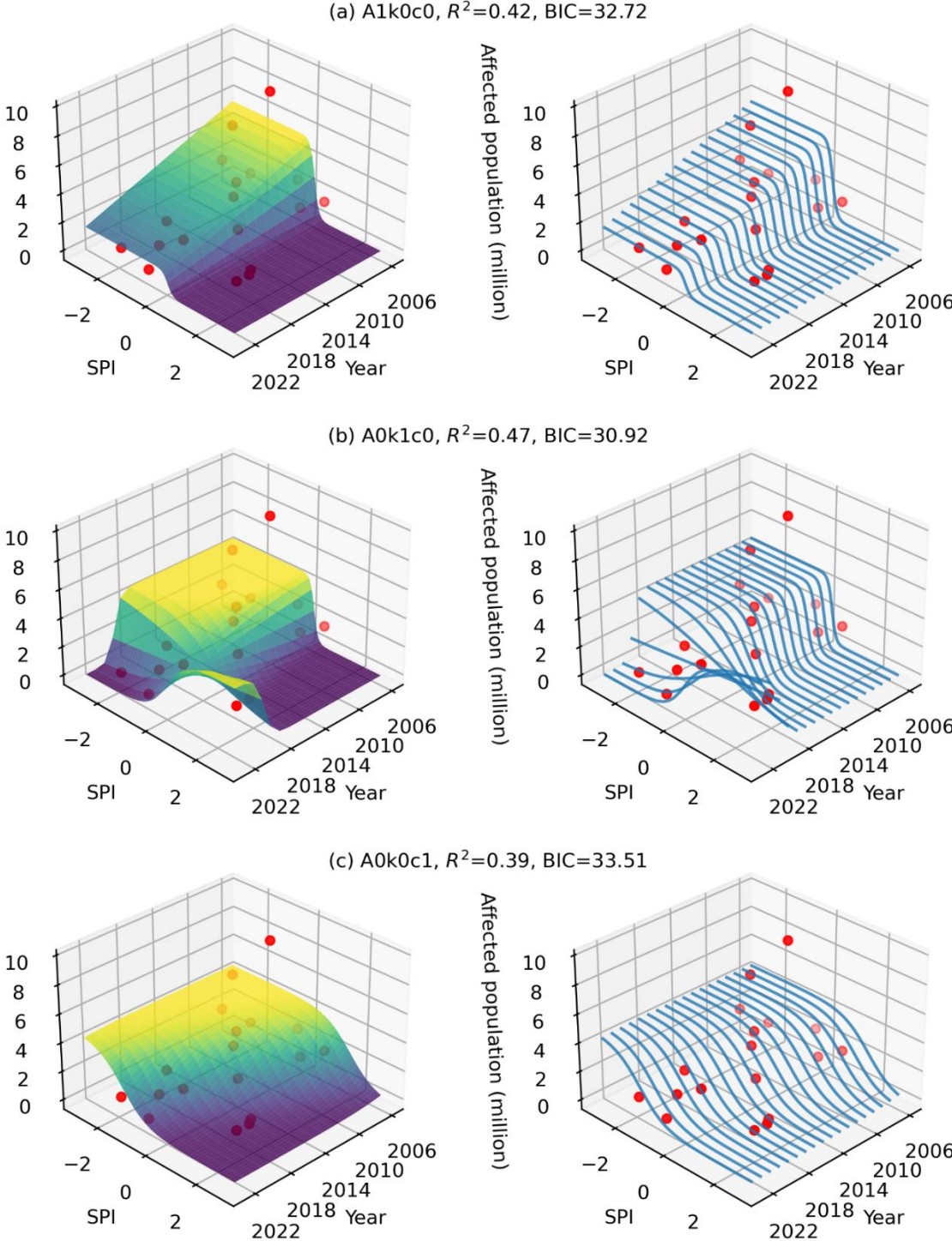

**Figure 7. Surface plots (left) and wireframe plots (right) for the nonstationary logistic functions (a) A1k0c0, (b) A0k1c0 and (c) A0k0c1 relating the drought-affected population to SPI and time for Yunnan Province.**

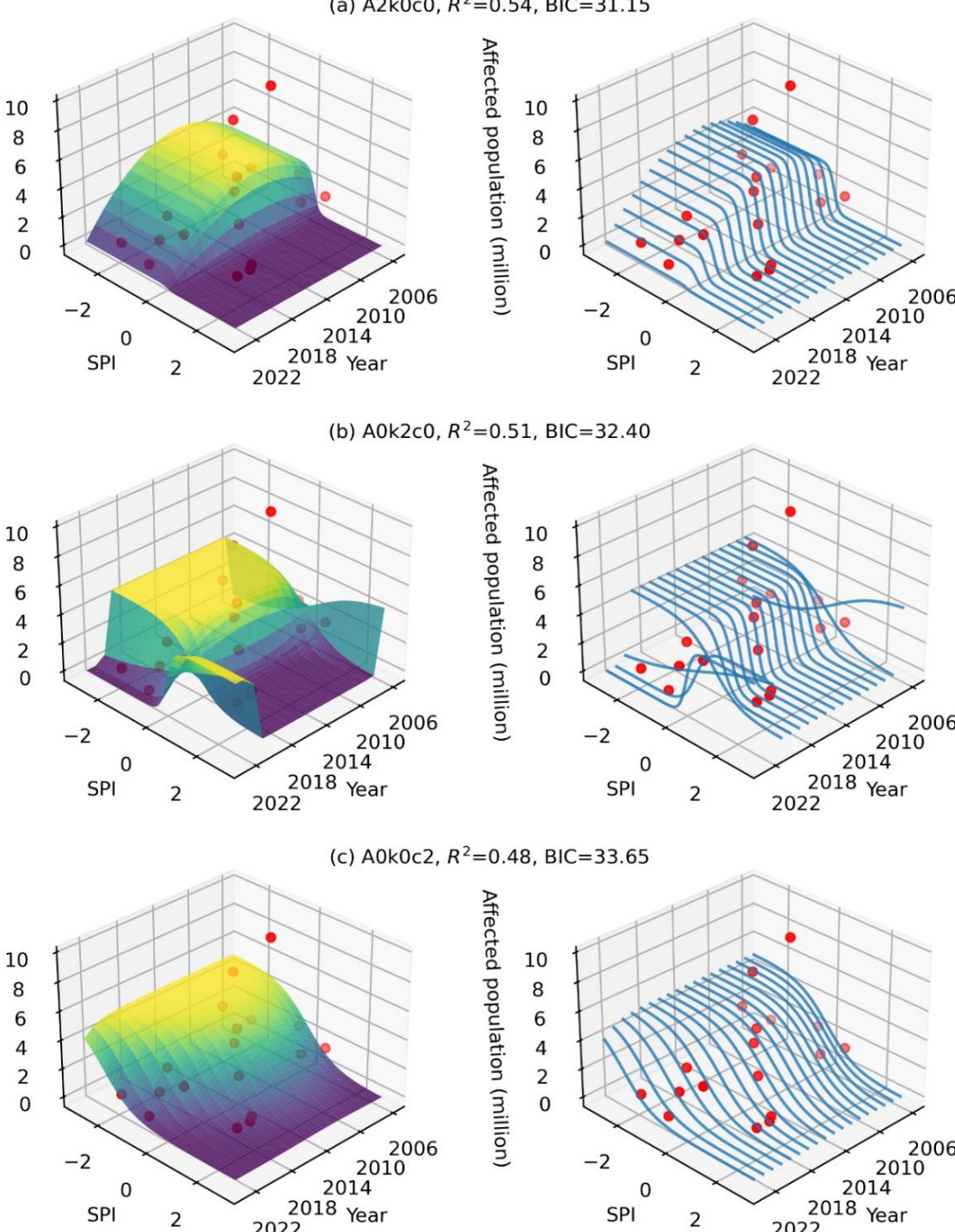

Figure 8. As for Figure 7, but for the nonstationary logistic functions (a) A2k0c0, (b) A0k2c0 and (c) A0k0c2.

## 4.3 Increasing drought-affected population

The stationary and nonstationary logistic functions are furthermore applied to Guangdong Province (Shao et al., 2020). Since the population of Guangdong is concentrated on the Pearl River Delta, recent years have witnessed serious water scarcity due to upstream reservoir impoundments and estuary saltwater intrusion (Weng et al., 2024).

From Figure 9, it is observed that there can be considerable drought-affected population when the precipitation is above average. The stationary logistic function A0k0c0 tends to capture the decrease of drought-affected population with SPI. Meanwhile, it is difficult for this function to capture the data points with high drought-affected population.

From Figure 10, it is seen that the three non-stationary logistic functions A1k0c0, A0k1c0 and A0k0c1 are more effective in characterising the dependency of drought-affected population on SPI and time. The linear function plays different parts in these three functions:

1) The linear magnitude parameter in A1k0c0 tends to fit the increase by enlarging the maximum drought loss by year. As shown in Figure 10a, it tends to capture the maximum drought-affected population of 1.50 million in 2020 and the second maximum drought-affected population of 1.24 million in 2021.

2) The linear shape parameter in A0k1c0 is observed to fit the observation data by changing the shape of response surface by year. As shown in Figure 10b, although the affected population in 2020 and 2021 is to some extent characterized, drought-affected population is seen to surprisingly increase with SPI in 2006. These results highlight the role that the shape parameter plays in determining the growth (reduction) rate.

3) The linear location parameter in A0k0c1 is shown to fit the observation data by fixing the maximum drought loss but shifting the response curves by year. As shown in Figure 10c, it tends to characterize the maximum and second maximum drought-affected population in recent years but does not seem to be as effective in characterizing drought-affected population in early years.

From Figure 11, it is observed that the three non-stationary logistic functions A2k0c0, A0k2c0 and A0k0c2 also tend to capture the drought-affected population. The result in Figure 11a is generally hard to interpret since the drought-affected population tends to initially decrease but then increase with time under a given SPI below zero. The results Figures 11b and 11c are respectively similar to those in Figures 10b and 10c. The implication is that the linear function in A0k1c0 and A0k0c1 can be as effective as the quadratic function in A0k2c0 and A0k0c2.

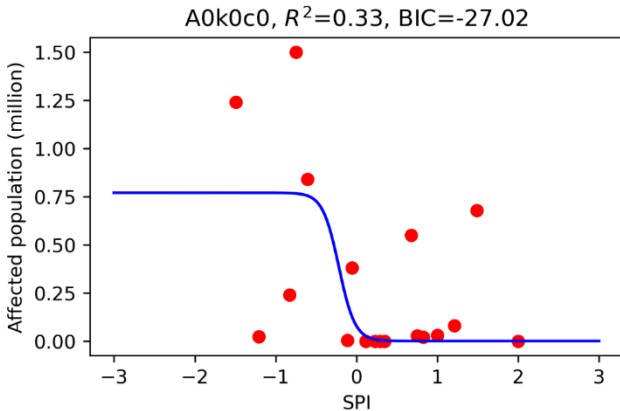

**Figure 9. Illustration of the stationary logistic function A0k0c0 fitting the relationship between SPI and drought-affected population for Guangdong Province.**

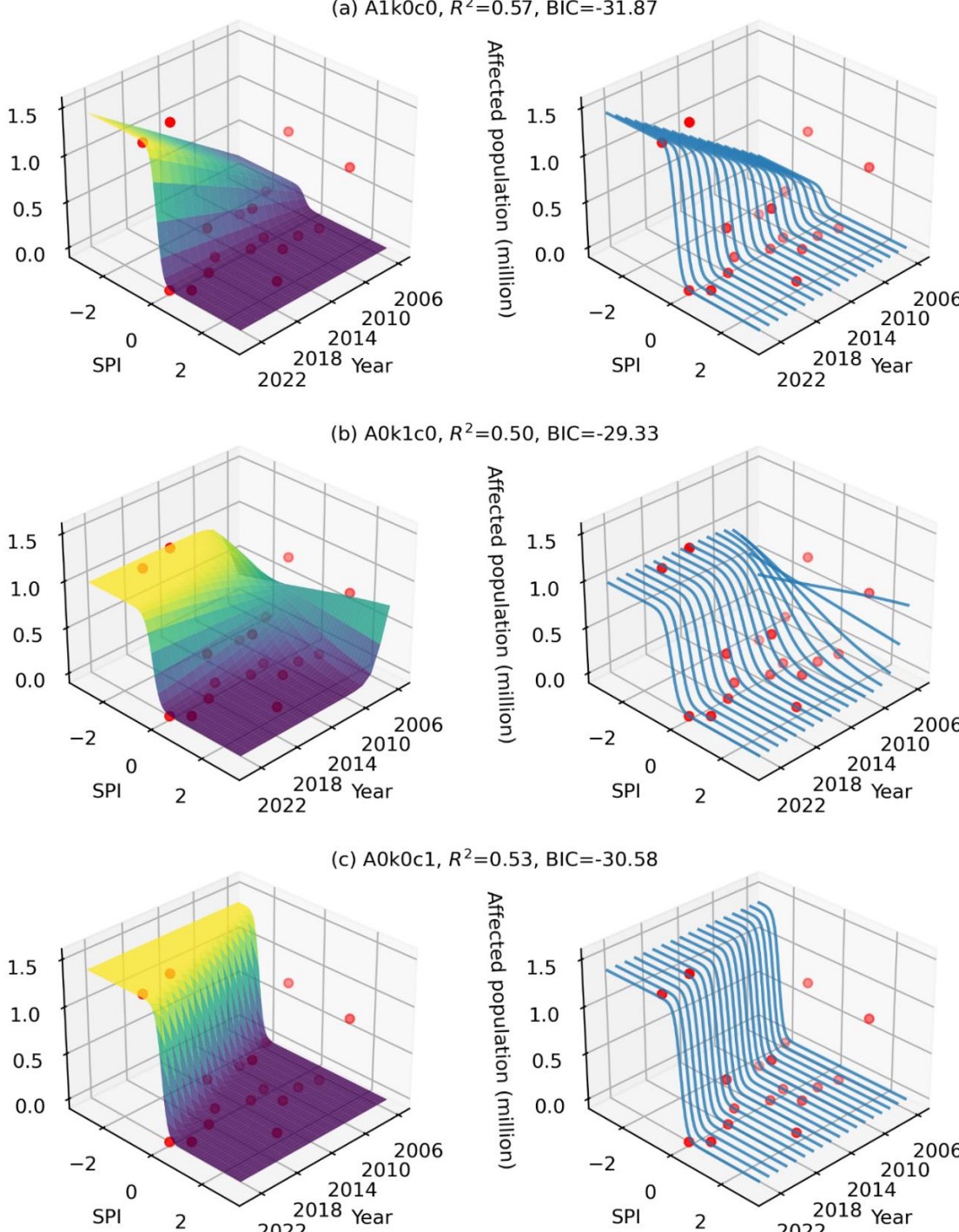

**Figure 10. Surface plots (left) and wireframe plots (right) for the nonstationary logistic functions (a) A1k0c0, (b) A0k1c0 and (c) A0k0c1 relating the drought-affected population to SPI and time for Guangdong Province.**

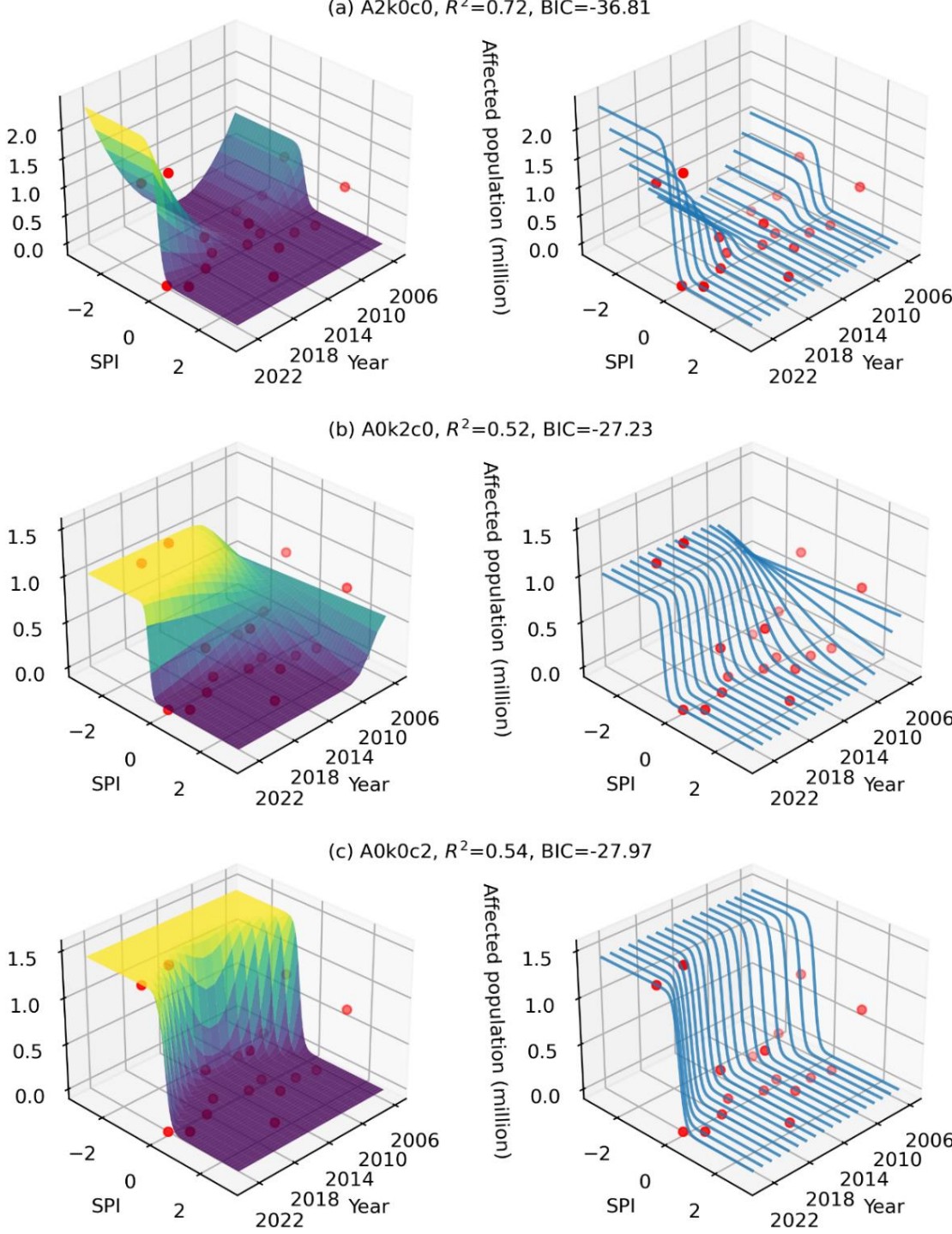

**Figure 11. As for Figure 10, but for the nonstationary logistic functions (a) A2k0c0, (b) A0k2c0 and (c) A0k0c2.**

**4.4 Goodness-of-fit**

The stationary and nonstationary logistic functions are set up to account for the drought-affected population based on the explanatory variables of time and SPI for 27 provincial administrative regions other than Beijing, Tianjin, Shanghai and Xizang. The $R^2$ for the three nonstationary logistic functions A1k0c0, A0k1c0 and A0k0c1 are plotted against that of linear regression based on time (Figure 12a) and also against that of the stationary logistic function (Figure 12b). The three scatter plots are generally above the 1:1 line. This result indicates that the consideration of time $t$ evidently enhances the proportion of total

variation explained by the non-stationary logistic functions. It is noted that the mean $R^2$ is respectively 0.307 for linear regression and 0.269 for the stationary logistic function A0k0c0. By contrast, the mean $R^2$ is respectively 0.512, 0.506 and 0.509 for A1k0c0, A0k1c0 and A0k0c1. Overall, the nonstationary logistic function A1k0c0 is of the highest $R^2$. This result highlights that the incorporation of time into the magnitude parameter can effectively deal with the non-stationary drought-affected population. Furthermore, the $R^2$ of the A2k0c0, A0k2c0 and A0k0c2 is investigated in Figure S15 of the

supplementary material. The results highlight the improvements in $R^2$ for the nonstationary logistic functions.

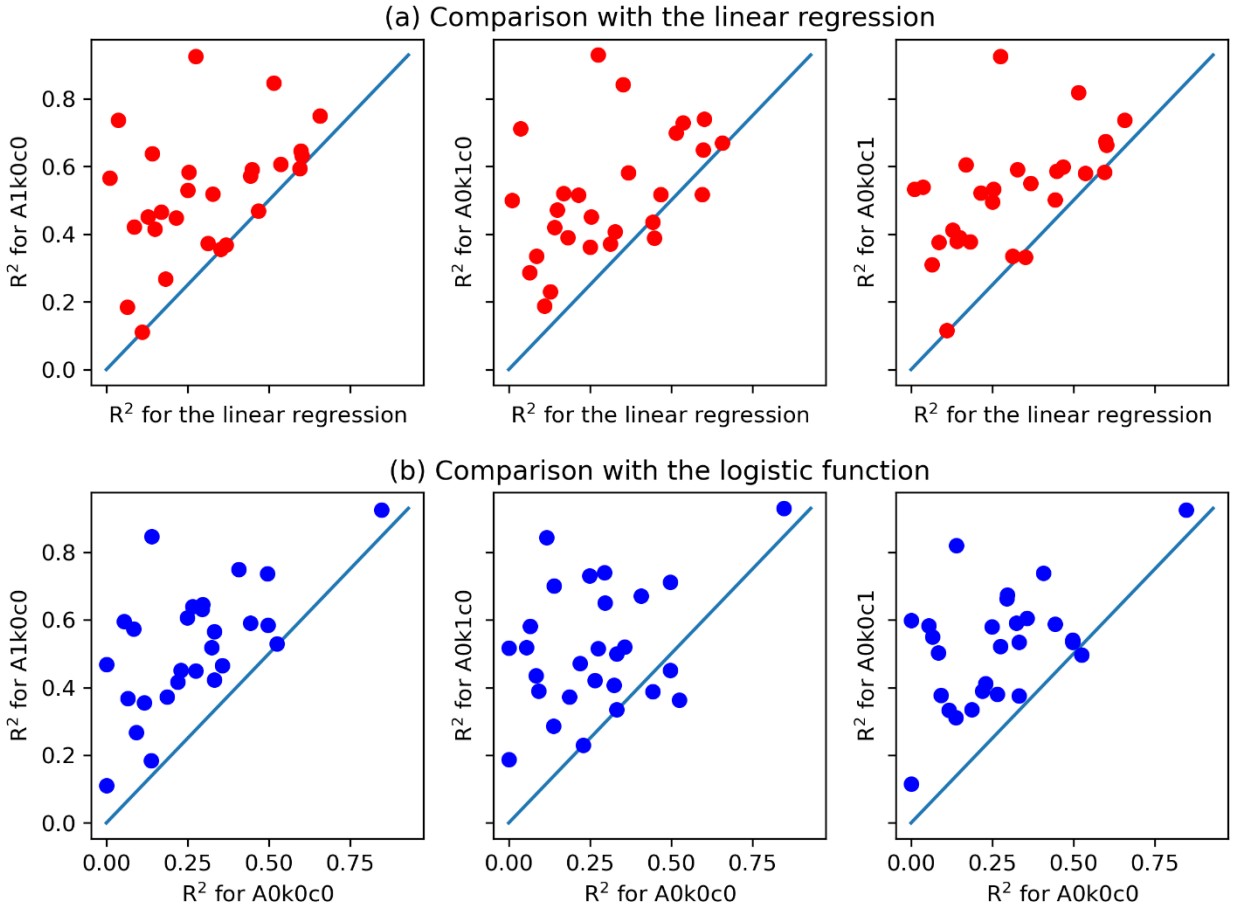

**Figure 12. Scatter plots of the $R^2$ for the three nonstationary logistic functions against the $R^2$ for (a) the linear regression and (b) the stationary logistic function A0k0c0.**

Furthermore, the BIC of the three nonstationary logistic functions A1k0c0, A0k1c0 and A0k0c1 is plotted against the BIC of the linear regression in Figure 13a and against that of the stationary logistic function in Figure 13b. Since the higher $R^2$ of the nonstationary logistic functions in Figure 12 is at the cost of an additional parameter (Neath and Cavanaugh, 2012), the BIC takes into account not only the number of parameters but also the mean squared error. It can be observed that the scatter plots in Figure 13 are largely below the 1:1 line. Considering that the BIC is a negatively oriented metric, this result suggests that there is a low risk of overfitting and that the information hidden in the significant correlation is deemed to be effectively exploited by the three non-stationary logistic functions. It is noted that the mean BIC is respectively –33.105 for linear regression and –29.365 for the stationary logistic function A0k0c0. By contrast, the mean BIC is respectively –34.980, –34.772 and –34.740 for A1k0c0, A0k1c0 and A0k0c1. As the nonstationary logistic function A1k0c0 is of the lowest BIC, it is highlighted that the incorporation of time into the magnitude parameter of the logistic function is effective in accounting for

the non-stationarity of drought losses. Furthermore, the BIC of the A2k0c0, A0k2c0 and A0k0c2 is investigated in Figure S16 of the supplementary material. Overall, the results are similar to these in Figure 13. The implication is that the incorporation of the linear function into the logistic function suffices to deal with the dependency of drought-affected population on SPI and time.

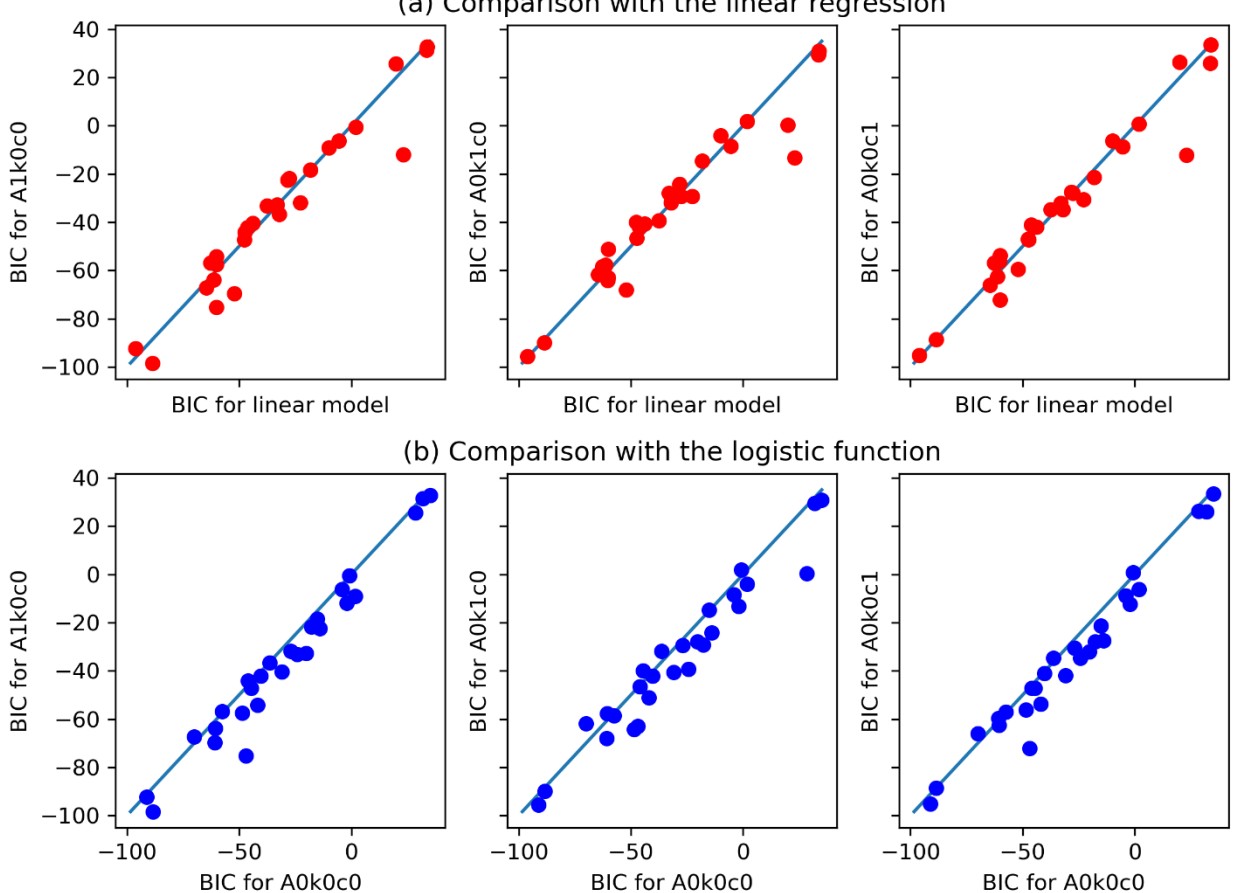

**Figure 13. As for Figure 12, but for the BIC.**

## 5 Discussion

This paper has furthermore designed experiments to investigate the robustness of the nonstationary logistic functions using the drought indices SPEI and scPDSI (AghaKouchak et al., 2021; Apurv and Cai, 2021; Zhao et al., 2024b). The additional results are presented in the supplementary material. Specifically, as for SPEI, the correlation is presented in Figure S1 and the plots for Yunnan and Guangdong Provinces in Figures S2 to S7; as for scPDSI, the correlation is presented in Figure S8 and the

plots for Yunnan and Guangdong Provinces in Figures S9 to S14. Overall, the results under SPEI and scPDSI conform to these under SPI. While the nonstationarity plays an important part in the relationship between drought-affected population and drought conditions, it is highlighted that the nonstationary logistic functions are effective in characterising the dependency of drought-affected population on drought conditions and time. In the meantime, it is pointed out that different drought indices are of varying efficiency in characterizing the drought conditions. For example, the lower $R^2$ in Figure 6 is largely due to the correspondence of maximum drought-affected population with average precipitation in the year 2010; the $R^2$ evidently increases from 0.22 under SPI (Figure 6) to 0.42 under SPEI (Figure S2), and furthermore to 0.58 under scPDSI (Figure S9). This result highlights that drought conditions depend on precipitation and also on other hydroclimatic variables like evapotranspiration, recharge and runoff (Wells et al., 2004; Yin et al., 2022b).

The nonstationary intensity loss functions developed in this paper complement existing studies on hydroclimatic processes of droughts (Garrido-Perez et al., 2024; Haile et al., 2020; Todisco et al., 2013). The frequency, duration and intensity are three important characteristics of drought (Baez-Villanueva et al., 2024; Entekhabi, 2023; Liu et al., 2024; Mishra and Singh, 2010; Yang et al., 2024). Given a threshold for the identification of drought events, the frequency is generally defined as the number of drought events in a certain period (one year for example), the duration as the timespan of a drought event and the intensity as the cumulative sum of the drought index (AghaKouchak et al., 2021; Chiang et al., 2021). Given that the SPI is derived for annual precipitation in this paper, the SPI values are expected to reflect the conditions of drought frequency, duration and intensity across different years. It is noted that the use of annual precipitation is mainly due to the fact that the drought-affected population by province is available at the annual timescale. It is possible that drought losses are available on an event scale. In that case, event-based analysis becomes feasible. That is, both drought loss and intensity can be quantified for each drought event; and then the effectiveness of the logistic function can be tested.

Focusing on drought indices such as SPI, PDSI, SPEI and SRI, previous studies have presented in-depth investigations about past changes and future projections of meteorological, hydrological, agricultural and socio-economic droughts (Apurv and Cai, 2021; Hao and Singh, 2015; Mishra and Singh, 2010). Under climate change, droughts are increasingly found to be interconnected with other extreme events including heatwaves (Yin et al., 2022a), tropical cyclones (Gao et al., 2024c), drought-flood abrupt alternation (Shi et al., 2021) and summer drought-flood coexistence (Wu et al., 2006). This paper proposes to incorporate time as a covariate to capture the overall trend of nonstationary drought losses. One remarkable feature of the proposed intensity loss function is the explicit estimation of drought loss under different combinations of drought indices and time. As the frequency and intensity of these compound disasters continue to increase, the socioeconomic losses are expected to rise in the future. The relationship between socioeconomic losses and other disaster indices can readily be investigated at local and regional scales. Given that the logistic function is already an established growth model in biosciences (Tsoularis and Wallace, 2002), it is expected that the proposed functions can be used to characterize the growth of drought loss with drought conditions characterized by different drought indices.

**6 Conclusions**

This paper has presented nonstationary intensity loss functions for drought impact assessment. On the one hand, the classic logistic function that has three parameters, i.e., magnitude, shape and location, presents a stationary formulation of the growth of drought losses with drought conditions. On the other hand, the incorporations of time as linear and quadratic functions into the magnitude, shape and location parameters facilitate in total six nonstationary logistic functions. A case study is presented for the drought-affected population by province in China during the period from 2006 to 2023. The results highlight that despite the fact that drought-affected population can either decrease or increase with time, the joint use of both SPI and time as explanatory variables leads to effective characterization of drought-affected population. In comparison with the stationary logistic function, the effectiveness of the nonstationary logistic functions is indicated not only by higher $R^2$, which indicates reasonable proportion of total explained variation, but also by lower BIC, which suggests low risk of overfitting. Among the nonstationary logistic functions, the function incorporating the linear function of time into the magnitude parameter generally outperform the others in terms of higher $R^2$, lower BIC and clearer physical meanings. In conclusion, the nonstationary intensity loss functions developed in this paper can improve our understanding and respond to drought risks in an era of rapid socio-economic and environmental change. Future research could further enhance this framework by incorporating additional socio-economic variables, to refine the model's predictive capabilities and support targeted mitigation strategies.

**Acknowledgments**

This research is supported by the National Natural Science Foundation of China (2023YFF0804900 and 52379033) and the Guangdong Provincial Department of Science and Technology (2019ZT08G090).

**CRediT authorship contribution statement**

Tongtiegang Zhao: Writing – original draft, Visualization, Software, Methodology, Conceptualization. Zecong Chen: Validation, Resources, Data curation. Yongyong Zhang: Investigation, Formal analysis. Bingyao Zhang: Visualization, Data curation. Yu Li: Validation, Data curation.

**Declaration of competing interest**

The authors declare that they have no known competing financial interests or personal relationships that could have appeared to influence the work reported in this paper.


## Data Availability Statement

The drought-affected population is available from the Ministry of Water Resources of China (http://www.mwr.gov.cn/sj/tjgb/zgshzhgb/). The CHIRPS precipitation data is available from the Climate Hazards Center at the University of California, Santa Barbara (https://www.chc.ucsb.edu/data/chirps). The SPEI data is available from the Global

SPEI database (https://spei.csic.es/database.html). The scPDSI data is available from the Climatic Research Unit (CRU) (https://crudata.uea.ac.uk/cru/data/drought/).

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
