# Peer review of "An extension of the logistic function to account for nonstationary drought losses"

_EGUsphere, 2024_

## Author Comment (AC1)

**Reviewer #1:**

*The relationship between socio-economic loss and drought intensity can be nonstationary, i.e., temporally changing, considering that economic growth can increase the exposure to droughts and that infrastructure developments can decrease the vulnerability to droughts. This study focused on an important issue that the response function of population to drought might be non-stationary. This work would provide rich information as references in guiding climate change mitigation.*

Thank you very much for the brief summary and the positive comments on the paper. Given that the intensity loss function plays a critical part in drought impact assessment, this paper presents an extension of the classic logistic function to account for temporal changes of drought losses.

*I have some minor suggestions for consideration.*

Thank you very much for the constructive comments. Accordingly, we have conducted a thorough revision of the whole paper.

Below please find the point-by-point responses.

*The authors attempted to explore the relationship between drought-affected population and drought intensity. I think it is better to say it is the population exposure, rather than loss. For the definition of loss, the readers might think it is economic damages or death.*

Thank you for the insightful comment. We have carefully considered the difference between "exposure" and "loss." While "exposure" typically refers to potential risk, the "drought-affected population" in our research is directly extracted from statistic data provided by the Ministry of Water Resources of China (*http://www.mwr.gov.cn/sj/tjgb/zgshzhgb/*). On its website, the data is noted to represent real occurrences and more closely aligns with the concept of "loss".

"The drought loss data is sourced from the Ministry of Water Resources (MWR) of China to test the stationary and nonstationary intensity loss functions. It is noted that the MWR has since 2006 published by year "Bulletin of Flood and Drought Disaster in China". The name of the bulletin was changed to "China Flood and Drought Disaster Prevention Bulletin" in 2019. By collating floods and droughts reported by provincial governments and river basin commissions, the MWR has presented in the bulletin major events of droughts and floods across the 31 provinces in mainland China. As to droughts and floods in each province, the bulletin provides by year the quantitative socio-economic losses, contingency plans and retrospective analysis of prevention and control measures.

The attention is paid to the drought-affected population, which represents the actual number of individuals suffering from drought events as recorded in official reports. In Figure 2 are the multi-annual mean drought-affected population, maximum annual drought-affected population, mean annual precipitation and total population. From Figures 2a and 2b, it can be observed that provinces in Southwest China, including Yunnan, Guizhou and Sichuan Provinces, tend to have the largest population suffering from drought. Particularly in 2010, 8.82 million people in Yunnan Province and 5.44 million people in Guizhou Province were struck by a record-breaking drought event induced by the persistently positive Madden-Julian Oscillation (Lü et al., 2012). On the other hand, it can be seen from Figures 2c and 2d that there is neither low precipitation nor large population in Southwest China. In general, the large drought-affected population in Yunnan and Sichuan Provinces is attributed to the Karst landscape, which features small storage capacity, high infiltration rate and fast groundwater flow (Wan et al., 2016)." (Page 8, Lines 165 to 179)

*The current Introduction section provided a good summary of the socio-economic impacts of droughts. However, the story about socioeconomic exposure and loss is not clear. Most of previous studies focused on exposure only, very few studies quantified the economic loss (e.g., https://doi.org/10.1073/pnas.1802129115).*

Thank you for the constructive comment. The issue of socio-economic losses has been illustrated in the Introduction and Methods.

"Socio-economic losses are an integral part of droughts in environment management (AghaKouchak et al., 2021; Hoerling et al., 2014; Van Dijk et al., 2013). Although there exist extensive studies on hydroclimatic processes associated with droughts (Entekhabi, 2023; Mishra and Singh, 2010; Wang et al., 2023b; Yang et al., 2024; Zhang et al., 2021), far less attention is paid to socio-economic impacts of droughts (AghaKouchak et al., 2021; Apurv and Cai, 2021; Su et al., 2018). One possible cause is the lack of socio-economic data on droughts (Su et al., 2018; Yang et al., 2024). On the one hand, in situ observations, satellite remote sensing and earth system models generate a vast amount of hydroclimatic data (Hersbach et al., 2020; Pradhan et al., 2022; Zhang et al., 2024, 2021; Zhao et al., 2024b). Plenty of spatial-temporal data facilitate drought investigations at catchment, regional, continental and global scales and in pentad, monthly, seasonal and annual time steps (Gao et al., 2024b; Ma et al., 2022; Wang et al., 2023a). On the other hand, there are limited data on socio-economic losses due to droughts (AghaKouchak et al., 2021). Usually, drought losses have to be collected from statistical yearbooks issued by local and central governments and from survey reports provided by international organizations and commercial services (Chen et al., 2015; Hou et al., 2019)." (Page 3, Lines 57 to 68)

"There are socio-economic factors contributing to temporal changes, i.e., nonstationarity, of the intensity loss function (AghaKouchak et al., 2021; Chiang et al., 2021; Long et al., 2020). Firstly, the exposure to drought can increase with time owing to increases of population, accumulations of wealth and developments of infrastructure.

Secondly, the vulnerability under a given level of drought intensity may decrease with time considering engineering measures, such as constructions of water storage reservoirs and inter-basin water diversion projects. Thirdly, the resilience to drought can be improved by drought management measures such as sub-seasonal to seasonal hydroclimatic forecasting and forecast-informed reservoir operation. In general, the relationship between drought loss and intensity tends to evolve as time progresses due to socio-economic developments and deployments of engineering and non-engineering drought-coping strategies (Hou et al., 2019; Jonkman et al., 2008; Su et al., 2018)."
(Page 6, Lines 137 to 145)

*In the Method section, the authors showed many drought indices, such as SPEI, PDSI. But the authors only used the SPI. I would recommend only showing the drought indices in Introduction. Moreover, many recent studies have employed the TWS-DSI in exploring the drought events. For example, the following references show some applications of TWS-DSI. https://doi.org/10.1007/s11430-021-9927-x*

Thank you for the constructive suggestion. We have performed additional experiments on SPEI and scPDSI:

"This paper has furthermore designed experiments to investigate the robustness of the nonstationary logistic functions using the drought indices SPEI and scPDSI (AghaKouchak et al., 2021; Apurv and Cai, 2021; Zhao et al., 2024b). The additional results are presented in the supplementary material. Specifically, as for SPEI, the correlation is presented in Figure S1 and the plots for Yunnan and Guangdong Provinces in Figures S2 to S7; as for scPDSI, the correlation is presented in Figure S8 and the plots for Yunnan and Guangdong Provinces in Figures S9 to S14. Overall, the results under SPEI and scPDSI conform to these under SPI. While the nonstationary plays an important part in the relationship between drought-affected population and drought conditions, it is highlighted that the nonstationary logistic functions are effective in characterising the dependency of drought-affected population on drought conditions and time. In the meantime, it is pointed out that different drought indices are of varying efficiency in characterizing the drought conditions. For example, the lower $R^2$ in Figure 6 is largely due to the correspondence of maximum drought-affected population with average precipitation in the year 2010; the $R^2$ evidently increases from 0.22 under SPI (Figure 6) to 0.42 under SPEI (Figure S2), and furthermore to 0.58 under scPDSI (Figure S9). This result highlight that drought conditions depend on precipitation and also on other hydroclimatic variables like evapotranspiration, recharge and runoff (Wells et al., 2004; Yin et al., 2022b)." (Pages 24 to 25, Lines 365 to 377)

*In the Abstract and Results, the authors provide significant correlation of population and time/other factors. I would recommend providing the p-value to test its significance level.*

Thank you for your suggestion. The p-value has been added to the bar plots:

"The Pearson's correlation coefficient between drought-affected population and time as well as SPI are illustrated by bar plots in Figure 3. There are in total 31 provincial administrative regions in mainland China. Beijing, Tianjin, Shanghai and Xizang are not considered since they are free from drought-affected population in most years. This outcome is mainly due to ample water availability and water supply facilities (Long et al., 2020; Sun et al., 2021). For the other 27 provincial administrative regions, it can be observed from Figure 3a that the correlation coefficient between drought-affected population and time is mostly significantly negative. Meanwhile, it is slightly positive in Guangdong and Fujian Provinces although not significant. The implication is that the drought-affected population mostly exhibits a decreasing trend as time progresses and sometimes shows an increasing trend. From Figure 3b, it is seen that the correlation coefficient between drought-affected population and SPI is in general significantly negative. This result suggests that drought-affected population tends to decrease as the amount of precipitation increases. Overall, the correlation coefficients in Figure 3 point out that it is reasonable to use both time and SPI as explanatory variables of drought-affected population." (Page 11, Lines 217 to 227)

[Figure]

[Figure]

Figure 3. Correlation coefficient between drought-affected population and (a) time as well as (b) SPI by province. Alongside the bars are *, ** and *** respectively indicating the significance at the levels of 0.10, 0.05 and 0.01. Bars without * imply non-significant correlation coefficients.

*In Figs. 4-5, I would recommend providing a statistical significance test.*

Thank you for your suggestion. The p-value has been added to the plots.

[Figure]

Figure 4. Scatter plots of drought-affected population against (a) time and (b) SPI in Yunnan Province.

[Figure]

Figure 5. As for Figure 4, but for Guangdong Province." (Page 12, Lines 236 to 246)

*I would recommend omitting the term 'novel' across the manuscript.*

Thank you for the suggestion. Accordingly, the word "novel" has been removed from the whole paper.

*The writing quality can be improved. For example, "To examine the effectiveness, a case study is devised for the…" I would recommend using 'designed', rather than 'devised'.*

Thank you for the suggestion. We have polished and proofread the whole paper.

*Climate change impacts on hydrological cycle and droughts have received growing attention. Could you briefly introduce some physical mechanism behind the drought evolution in mainland China? How about extending this framework to compound hazards?*

Thank you very much for the constructive comment. Climate change impacts are illustrated in the discussion:

"Under climate change, droughts are increasingly found to be interconnected with other extreme events including heatwaves (Yin et al., 2022a), tropical cyclones (Gao et al., 2024c), drought-flood abrupt alternation (Shi et al., 2021) and summer drought-flood coexistence (Wu et al., 2006). This paper proposes to time as a covariate to capture the

overall trend of nonstationary drought losses. One remarkable feature of the proposed intensity loss function is the explicit estimation of drought loss under different combinations of drought indices and time. As the frequency and intensity of these compound disasters continue to increase, the socioeconomic losses are expected to rise in the future. The relationship between socioeconomic losses and other disaster indices can readily be investigated at local and regional scales. Given that the logistic function is already an established growth model in biosciences (Tsoularis and Wallace, 2002), it is expected that the proposed functions can be used to characterize the growth of drought loss with drought conditions characterized by different drought indices." (Page 25, Lines 391 to 400)

---

## Author Comment (AC2)

**Reviewer #2:**

*The main research content of this paper focuses on the nonstationary relationship between drought losses and drought intensity, proposing nonstationary intensity-loss function based on the Logistic function.*

We appreciate the brief summary of the paper. Considering that the intensity loss function plays a critical part in drought impact assessment. This paper presents an extension of the classic logistic function to account for temporal changes of drought losses.

*If the authors can address the following issues, this manuscript has the potential to be accepted:*

We are grateful to you for the insightful and constructive comments. Accordingly, we have conducted a thorough revision.

Below please find the point-by-point responses.

1. *The abstract provides a concise summary of the study; however, it could benefit from a more explicit mention of the key findings and their significance. Consider highlighting the specific improvements your model offers over existing approaches and the practical implications of these improvements.*

Thank you for your valuable suggestion. We have revised the abstract to explicitly highlight the key findings and their significance:

"While the stationary intensity loss function is fundamental to drought impact assessment, the relationship between drought loss and intensity can be nonstationary, i.e., changing as time progresses, owing to socio-economic developments. This paper addresses this critical gap by modelling nonstationary drought losses. Specifically, the time is explicitly formulated by linear and quadratic functions and then incorporated into the magnitude, shape and location parameters of the logistic function to derive in total six nonstationary intensity loss functions. To examine the effectiveness, a case study is designed for the drought-affected population by province in mainland China during the period from 2006 to 2023. The results highlight the existence of nonstationarity in that the drought-affected population exhibits significant correlation not only with standard precipitation index but also with time. The proposed nonstationary intensity loss functions are shown to outperform not only the classic logistic function but also the linear regression. They present effective characterizations of observed drought loss in different ways: 1) the nonstationary function with the flexible magnitude parameter fits the data by adjusting the maximum drought loss by

year; 2) the nonstationary function with the flexible shape parameter works by modifying the growth rate of drought loss with intensity; and 3) the nonstationary function with the flexible location parameter acts by shifting the response curves along the axis by year. Among the nonstationary logistic functions, the function incorporating the linear function of time into the magnitude parameter generally outperform the others in terms of high coefficient of determination, low Bayesian information criterion and explicit physical meaning. Taken together, the nonstationary intensity loss functions developed in this paper can serve as an effective tool for drought management." (Page 2, Lines 21 to 36)

2. *In the introduction, this manuscript positions itself as addressing the nonstationarity of drought losses, it could benefit from a more explicit comparison with existing approaches. For instance, what specific limitations of traditional logistic functions or linear regression does this study overcome? The literature review could be expanded to include recent studies on the topic. This would provide a clearer picture of the current state of the field and where your work fits within it.*

Thank you very much for the constructive suggestion. Upon the classic logistic function, the developments of the nonstationary intensity-loss function are detailed in the methods:

"**2.3 Stationary and non-stationary formulations**

There are socio-economic factors contributing to temporal changes, i.e., nonstationarity, of the intensity loss function (AghaKouchak et al., 2021; Chiang et al., 2021; Long et al., 2020). Firstly, the exposure to drought can increase with time owing to increases of population, accumulations of wealth and developments of infrastructure. Secondly, the vulnerability under a given level of drought intensity may decrease with time considering engineering measures, such as constructions of water storage reservoirs and inter-basin water diversion projects. Thirdly, the resilience to drought can be improved by drought management measures such as sub-seasonal to seasonal hydroclimatic forecasting and forecast-informed reservoir operation. In general, the relationship between drought loss and intensity tends to evolve as time progresses due to socio-economic developments and deployments of engineering and non-engineering drought-coping strategies (Hou et al., 2019; Jonkman et al., 2008; Su et al., 2018).

Without considering temporal changes, there is a stationary logistic function $L_{A0k0c0}(\cdot)$:

$$L_{A0k0c0}(SPI_t) = \frac{A_0}{1 + e^{k_0(SPI_t - c_0)}} \qquad (8)$$

To account for temporal change, the linear function that takes time $t$ as an explanatory variable (Cheng et al., 2014; Xiong et al., 2015) can be formulated for the parameters $A$, $k$ and $c$:

$$\begin{cases} A_t = A_0 + A_1 \times t \\ k_t = k_0 + k_1 \times t \\ c_t = c_0 + c_1 \times t \end{cases} \qquad (9)$$

in which $A_0$, $k_0$ and $c_0$ are the intercepts while $A_1$, $k_1$ and $c_1$ are the slopes. The incorporation of Eq. (9) into Eq. (8) yields the following three equations:

$$\begin{cases} L_{A1k0c0}(SPI_t) = \dfrac{A_0 + A_1 \times t}{1 + e^{k_0(SPI_t - c_0)}} \\ L_{A0k1c0}(SPI_t) = \dfrac{A_0}{1 + e^{(k_0 + k_1 \times t) \times (SPI_t - c_0)}} \\ L_{A0k0c1}(SPI_t) = \dfrac{A_0}{1 + e^{k_0(SPI_t - (c_0 + c_1 \times t))}} \end{cases} \qquad (10)$$

in which the logistic functions $L_{A0k1c0}(\cdot)$, $L_{A0k1c0}(SPI_t)$ and $L_{A0k0c1}(SPI_t)$ respectively have nonstationary magnitude, shape and location parameters.

Furthermore, the quadratic function can be used to accommodate possibly nonlinear changes:

$$\begin{cases} A_t = A_0 + A_1 \times t + A_2 \times t^2 \\ k_t = k_0 + k_1 \times t + k_2 \times t^2 \\ c_t = c_0 + c_1 \times t + c_2 \times t^2 \end{cases} \qquad (11)$$

The incorporation of Eq. (11) into Eq. (8) yields another three equations:

$$\begin{cases} L_{A2k0c0}(SPI_t) = \dfrac{A_0 + A_1 \times t + A_2 \times t^2}{1 + e^{k_0(SPI_t - c_0)}} \\ L_{A0k2c0}(SPI_t) = \dfrac{A_0}{1 + e^{(k_0 + k_1 \times t + k_2 \times t^2) \times (SPI_t - c_0)}} \\ L_{A0k0c2}(SPI_t) = \dfrac{A_0}{1 + e^{k_0(SPI_t - (c_0 + c_1 \times t + c_2 \times t^2))}} \end{cases} \qquad (12)$$

In Eq. (8), Eq. (10) and Eq. (12), the subscripts "Ax", "kx" and "cx" are respectively for the magnitude, shape and location parameters. As to "x", the values 0, 1 and 2 respectively indicate the non-involvement of time, the linear function of time and the quadratic function of time. As a result, the logistic function is non-stationary when x is 1 or 2. For example, $L_{A1k0c0}(SPI_t)$ represents the nonstationary logistic function involving the linear function of time for the magnitude parameter.

The fitting of the stationary and nonstationary functions is considered to be a nonlinear least-squares problem by searching for the set of parameters that minimize the sum of squares of residuals. It is performed by the curve_fit function in the SciPy optimization toolbox (Virtanen et al., 2020)." (Pages 6 to 8, Lines 136 to 161)

3. *The assumption of linear trends for the magnitude, shape, and location parameters may oversimplify the complex socio-economic influencing drought losses. It would*

*strengthen the methodology to either justify this assumption or explore the feasibility of incorporating nonlinear trends.*

Thank you for the constructive suggestion. As we have used both linear and quadratic functions to account for possible nonstationary relationships, we have elaborated on the results under both linear and quadratic functions:

[revised manuscript text omitted]

" (Pages 13 to 21, Lines 248 to 328)

4. *This manuscript mentions that changes in the disaster-affected population are related to socio-economic development. However, further details on how specific*

*socio-economic changes (e.g., population growth, infrastructure development) affect drought losses could be elaborated. This would help to better illustrate the importance of the study. For example, the author should be attempted to include socio-economic variables together with SPI as model inputs to further enhance the model's explanatory power, instead of just considering time as a covariate.*

Thank you very much for the insightful suggestions. We agree on the importance of socio-economic variables. It is noted that as a critical explanatory variable, the time can be considered to be a proxy of socio-economic variables. The importance is illustrated in the methods:

"There are socio-economic factors contributing to temporal changes, i.e., nonstationarity, of the intensity loss function (AghaKouchak et al., 2021; Chiang et al., 2021; Long et al., 2020). Firstly, the exposure to drought can increase with time owing to increases of population, accumulations of wealth and developments of infrastructure. Secondly, the vulnerability under a given level of drought intensity may decrease with time considering engineering measures, such as constructions of water storage reservoirs and inter-basin water diversion projects. Thirdly, the resilience to drought can be improved by drought management measures such as sub-seasonal to seasonal hydroclimatic forecasting and forecast-informed reservoir operation. In general, the relationship between drought loss and intensity tends to evolve as time progresses due to socio-economic developments and deployments of engineering and non-engineering drought-coping strategies (Hou et al., 2019; Jonkman et al., 2008; Su et al., 2018)." (Page 6, Lines 137 to 145)

5. *The analysis effectively highlights differences between regions such as Yunnan and Guangdong. However, the underlying drivers of these variations (e.g., climate conditions, socio-economic factors) are insufficiently explored. Adding a discussion on the interaction between climate and socio-economic variables would enrich the interpretation.*

Thank you very much. For these two provinces, we have designed new experiments to test the robustness of the findings and the importance of drought indices:

"This paper has furthermore designed experiments to investigate the robustness of the nonstationary logistic functions using the drought indices SPEI and scPDSI (AghaKouchak et al., 2021; Apurv and Cai, 2021; Zhao et al., 2024b). The additional results are presented in the supplementary material. Specifically, as for SPEI, the correlation is presented in Figure S1 and the plots for Yunnan and Guangdong Provinces in Figures S2 to S7; as for scPDSI, the correlation is presented in Figure S8 and the plots for Yunnan and Guangdong Provinces in Figures S9 to S14. Overall, the results under SPEI and scPDSI conform to these under SPI. While the nonstationary plays an important part in the relationship between drought-affected population and drought conditions, it is highlighted that the nonstationary logistic functions are effective in characterising the dependency of drought-affected population on drought conditions

and time. In the meantime, it is pointed out that different drought indices are of varying efficiency in characterizing the drought conditions. For example, the lower $R^2$ in Figure 6 is largely due to the correspondence of maximum drought-affected population with average precipitation in the year 2010; the $R^2$ evidently increases from 0.22 under SPI (Figure 6) to 0.42 under SPEI (Figure S2), and furthermore to 0.58 under scPDSI (Figure S9). This result highlight that drought conditions depend on precipitation and also on other hydroclimatic variables like evapotranspiration, recharge and runoff (Wells et al., 2004; Yin et al., 2022b)." (Pages 24 to 25, Lines 365 to 377)

6. *The discussion should not merely restate the results but should also delve deeper into the interpretation of these results in the context of existing literature. The discussion could better position the proposed nonstationary models within the broader literature on drought loss modeling. For instance, how do the findings align with or differ from previous studies, such as those using alternative loss functions or indices?*

Thank you very much for the constructive comment. The section of discussion has been improved by incorporating the results of additional experiments and the findings of peer studies:

"**5. Discussion**

[revised manuscript text omitted]

8. *Figures 6 and 7 provide valuable insights, but the 3D surface plots may not be intuitive for all readers. Supplementary 2D plots or contour maps could improve accessibility while maintaining scientific rigor.*

Thank you very much for the valuable suggestion. We have generated 2D heatmaps for the 3D surface plots and presented them in the supplementary material:

[Figure]

Figure S17. 2D heatmaps for for the nonstationary logistic functions (a) A1k0c0, (b) A0k1c0 and (c) A0k0c1 relating the drought-affected population to time and SPI for Yunnan Province.

[Figure]

Figure S18. As for Figure S17, but for Guangdong Province.